# Palovarotene reduces heterotopic ossification in juvenile FOP mice but exhibits pronounced skeletal toxicity

John B Lees-Shepard, Sarah-Anne E Nicholas, Sean J Stoessel,
Parvathi M Devarakonda, Michael J Schneider, Masakazu Yamamoto,
David J Goldhamer*

Department of Molecular and Cell Biology, University of Connecticut Stem Cell Institute, University of Connecticut, Storrs, United States

**Abstract** Fibrodysplasia ossificans progressiva (FOP) is a rare genetic disorder characterized by debilitating heterotopic ossification (HO). The retinoic acid receptor gamma agonist, palovarotene, and antibody-mediated activin A blockade have entered human clinical trials, but how these therapeutic modalities affect the behavior of pathogenic fibro/adipogenic progenitors (FAPs) is unclear. Using live-animal luminescence imaging, we show that transplanted pathogenic FAPs undergo rapid initial expansion, with peak number strongly correlating with HO severity. Palovarotene significantly reduced expansion of pathogenic FAPs, but was less effective than activin A inhibition, which restored wild-type population growth dynamics to FAPs. Palovarotene pretreatment did not reduce FAPs' skeletogenic potential, indicating that efficacy requires chronic administration. Although palovarotene inhibited chondrogenic differentiation in vitro and reduced HO in juvenile FOP mice, daily dosing resulted in aggressive synovial joint overgrowth and long bone growth plate ablation. These results highlight the challenge of inhibiting pathological bone formation prior to skeletal maturation.
DOI: https://doi.org/10.7554/eLife.40814.001

*For correspondence:
david.goldhamer@uconn.edu

## Introduction

Fibrodysplasia ossificans progressiva (FOP) is a rare genetic disease in which extraskeletal bone forms in skeletal muscle, tendon, and the associated soft connective tissues. In FOP, heterotopic ossification (HO) results from dysregulated signaling through the type one bone morphogenetic protein (BMP) receptor ACVR1 (ALK2) (*Shore et al., 2006*). Whereas several *ACVR1* mutations have been identified in FOP patients, the most prevalent is a point mutation that results in an arginine to histidine substitution at position 206 of the ACVR1 receptor [ACVR1(R206H)] (*Shore et al., 2006*). This amino acid change, which is within the cytoplasmic glycine-serine domain, upstream of the serine/threonine kinase domain, renders the receptor hyperactive to BMP ligands (*Billings et al., 2008*; *Hatsell et al., 2015*; *Hino et al., 2015*; *Haupt et al., 2018*) and confers novel responsiveness to activin ligands (*Hatsell et al., 2015*; *Hino et al., 2015*). With an appropriate physiological trigger, this altered signaling inappropriately activates the osteogenic program in tissue-resident progenitors, ultimately leading to endochondral HO. Although muscle injury and inflammation are strong triggers for 'flares' leading to HO, HO lesions often develop without a known trigger (commonly referred to as spontaneous HO). Progressive episodes of spontaneous HO generally begin in early childhood and increase in frequency and severity during childhood and adolescence (*Pignolo et al., 2018*; *Pignolo et al., 2016*). In individuals with FOP, significant HO-related disability occurs prior to skeletal maturity (*Pignolo et al., 2018*). Hence, it is important for FOP therapeutics to exhibit an acceptable safety profile in juvenile patients.

To facilitate drug discovery efforts and to investigate the cellular and physiological basis of FOP (*Lees-Shepard and Goldhamer, 2018*), we and others have recently developed conditional mouse genetic models of FOP (*Hatsell et al., 2015*; *Lees-Shepard et al., 2018*), which circumvent the perinatal lethality of constitutive $Acvr1^{R206H}$ mice (*Chakkalakal et al., 2012*). Using FOP mice, we identified fibro/adipogenic progenitors (FAPs), PDGFRα+ multipotent cells widely distributed in muscles and other tissues, as a key cell-of-origin of heterotopic cartilage and bone (*Lees-Shepard et al., 2018*). Targeting $Acvr1^{R206H}$ expression to FAPs results in robust injury-induced HO, and early-onset spontaneous HO in juvenile mice (*Lees-Shepard et al., 2018*). The current study more fully characterizes FAP-directed spontaneous HO, which shows marked similarities to the human condition.

FOP mice (*Hatsell et al., 2015*) and patient-derived induced pluripotent stem cells (*Hino et al., 2015*) were instrumental in the discovery of the fundamental and unexpected role of activin ligands in FOP pathogenesis, and antibody-based activin inhibition has emerged as a leading candidate therapeutic approach (*Hatsell et al., 2015*; *Lees-Shepard et al., 2018*; *Upadhyay et al., 2017*) that is now being evaluated in clinical trials. A second treatment modality, the retinoic acid receptor gamma (RARγ) agonist, palovarotene (*Chakkalakal et al., 2016*; *Shimono et al., 2011*; *Sinha et al., 2016*), has already shown some promise in clinical trials with adult FOP patients and enrollment is underway for safety and efficacy studies in children. RARγ agonists have been shown to dampen BMP signaling by reducing SMAD1/5/8 phosphorylation (*Shimono et al., 2011*), potentially by increasing proteasome-mediated SMAD degradation, as has been shown for all-*trans*-retinoic acid (*Sheng et al., 2010*). These effects likely explain, at least in part, the inhibitory effects of RARγ agonists on chondrogenic and osteogenic differentiation in BMP-induced and genetic models of HO (*Chakkalakal et al., 2016*; *Inubushi et al., 2018*; *Shimono et al., 2014*; *Shimono et al., 2011*; *Sinha et al., 2016*; *Wheatley et al., 2018*). Intriguingly, pretreating bone marrow-derived mesenchymal stem cells with the RARγ agonists CD1530 (*Shimono et al., 2011*) or NRX204647 (*Shimono et al., 2014*) blocked BMP2-induced skeletogenic differentiation, possibly by reprogramming these cells to a non-skeletal lineage (*Shimono et al., 2014*; *Shimono et al., 2011*). These latter results raise the possibility that short-term therapeutic intervention could have long-term efficacy, thereby minimizing the potential for retinoid-associated skeletal toxicity (*DiGiovanna, 2001*) in children, as noted previously (*Shimono et al., 2011*).

In the present study, the effects of daily palovarotene treatment on juvenile FOP mice were assessed, with a focus on body-wide spontaneous HO and skeletal development. We also used a transplantation model and live-animal imaging to quantify cell population dynamics of $Acvr1^{R206H}$-expressing FAPs (R206H-FAPs) prior to the formation of calcified bone. The effects of palovarotene treatment on cell engraftment and population expansion were compared to those of an anti-activin A blocking antibody (ActA-mAb). Finally, we tested whether palovarotene pretreatment renders R206H-FAPs refractory to skeletogenic differentiation.

## Results

### Targeting $Acvr1^{R206H}$ expression to FAPs models spontaneous HO in FOP

To evaluate the efficacy of the RARγ agonist palovarotene on a cell type demonstrably relevant to FOP, we used the previously described $Acvr1^{tnR206H}$ mouse genetic model (*Lees-Shepard et al., 2018*) and targeted expression of $Acvr1^{R206H}$ to FAPs using the Pdgfrα-Cre driver (*Roesch et al., 2008*). The eGFP Cre-dependent reporter allele, $R26^{NG}$ (*Yamamoto et al., 2009*), was included to confirm the specificity of recombination driven by Pdgfrα-Cre (*Lees-Shepard et al., 2018*). We have previously shown that Pdgfrα-Cre-driven recombination of the $Acvr1^{tnR206H}$ allele reliably results in FOP-like spontaneous HO and reduces survival by 6-weeks-of-age (*Lees-Shepard et al., 2018*). Here we conducted a detailed natural history of progressive HO and osteochondroma formation in this model, focusing on age of disease onset, rate of disease progression, and survival. These studies served as the foundation for subsequent drug testing.

$Acvr1^{tnR206H/+}$;$R26^{NG/+}$;Pdgfrα-Cre (Pdgfrα-R206H) mice were produced at Mendelian frequencies (n = 168 mice scored) and lacked overt abnormalities such as the great toe malformation that characterizes human FOP. Although HO was not detectable via gross observation at P14, whole-mount Alcian Blue and Alizarin Red staining revealed that two of ten mice exhibited early-stage HO

(*Figure 1A–F*; *Table 1*). These lesions were composed primarily of cartilage, with little calcification detectable (*Figure 1D–F*). Also evident were mild overgrowths of endogenous boney protuberances, particularly the deltoid tuberosity, which were observed in 40% of P14 Pdgfrα-R206H mice (*Figure 1C*; *Table 1*). These overgrowths were often stalk-like cartilage-capped projections that morphologically resemble the osteochondromas observed in FOP patients (*Deirmengian et al., 2008*; *Kaplan et al., 1993*; *Morales-Piga et al., 2015*).

In agreement with the natural course of human FOP (*Pignolo et al., 2018*; *Pignolo et al., 2016*), the incidence of HO in Pdgfrα-R206H mice increased with age and affected all regions and tissues

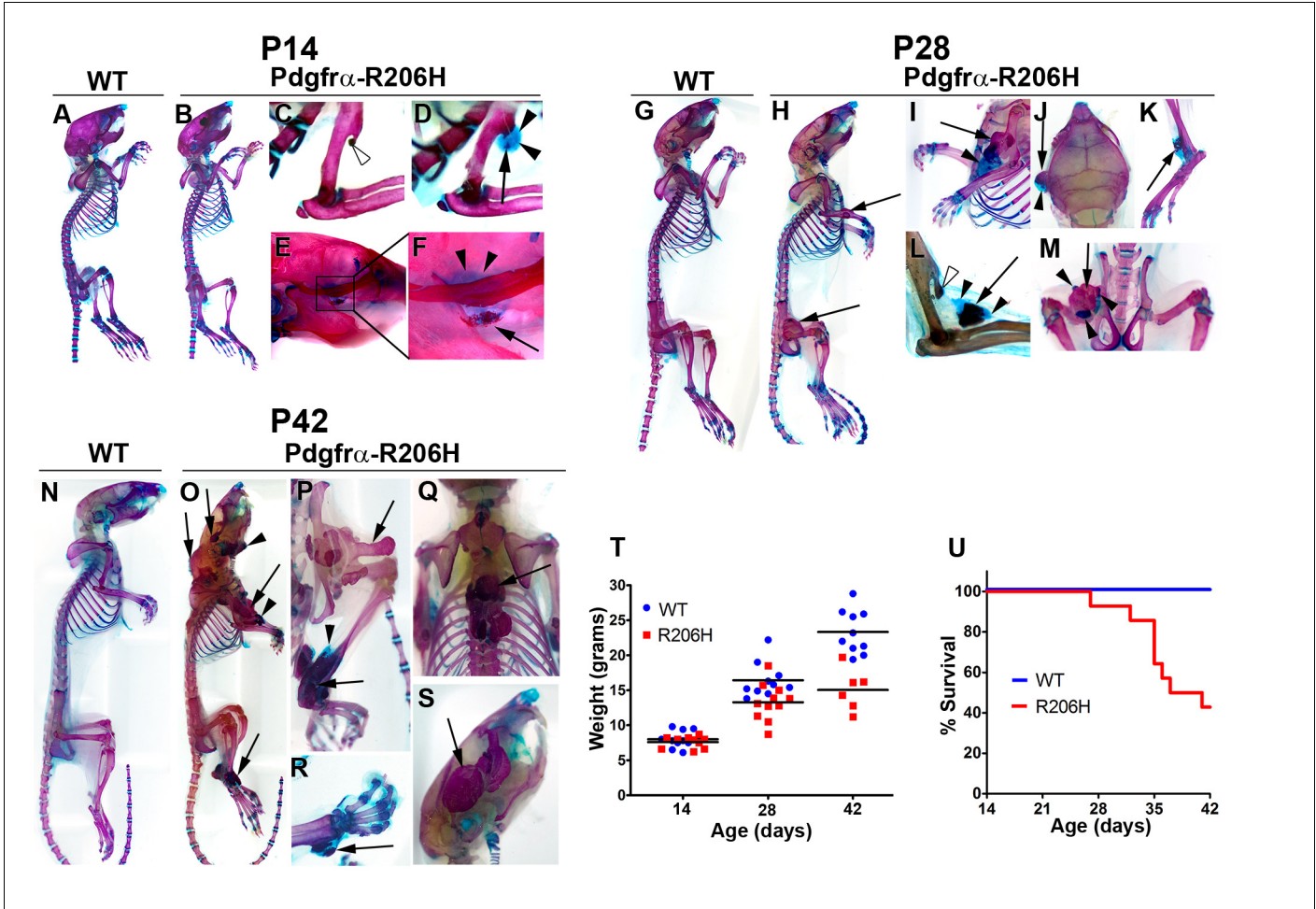

**Figure 1.** Natural history of juvenile Pdgfrα-R206H mice. (A–S) Whole mount skeletal preparations of wild-type (WT) and Pdgfrα-R206H mice stained with Alcian Blue and Alizarin Red (ABAR) to detect cartilage and bone, respectively. (A, G, N) HO was never observed in WT mice. (B) Pdgfrα-R206H mice rarely exhibited HO at P14. (C) A stalk-like osteochondroma (open arrowhead) emanating from the deltoid tuberosity of the humerus. (D, E, F) When present, HO at P14 was primarily comprised of cartilage (arrowheads) with minimal bone (arrow). (H) All P28 Pdgfrα-R206H mice presented multiple sites of HO (arrows). (I–M) Higher magnification images of HO in P28 Pdgfrα-R206H mice, which contained both boney (arrows) and cartilaginous (arrowheads) regions. (L) An osteochondroma (open arrowhead) and HO (arrows/arrowheads) in the same forelimb limb. (O–S) P42 Pdgfrα-R206H mice exhibited widespread boney HO (arrows), which occasionally contained peripheral cartilage (arrowheads). (P–S) Higher magnification images of HO in P42 Pdgfrα-R206H. (T) Compared to WT mice, Pdgfrα-R206H mice exhibited a 16% and 35% reduction in mean body weight at P28 (n = 11; p<0.01) and P42 (n = 6; p<0.0001), respectively. Means are depicted (black bars), and statistical significance was assessed by one-way ANOVA. (U) Pdgfrα-R206H mice survived to a median age of 39 days and exhibited a significantly reduced survival rate of 43% at P42, as assessed by log-rank (n = 14; p<0.0001).

DOI: https://doi.org/10.7554/eLife.40814.002

The following figure supplement is available for figure 1:

**Figure supplement 1.** A comparison of osteochondroma and HO formation in the Pdgfrα-R206H mouse forelimb.

DOI: https://doi.org/10.7554/eLife.40814.003

**Table 1.** Incidence of HO/overgrowth in untreated Pdgfrα-R206H mice

| Site | P14 (n = 10) | P28 (n = 11) | P42 (n = 6) |
|---|---|---|---|
| Ankle | 20% | 73% | 100% |
| Knee | 0% | 0% | 33% |
| Hip | 10% | 36% | 33% |
| Wrist | 0% | 45% | 66% |
| Abdomen | 0% | 0% | 0% |
| Forearm | 0% | 9% | 83% |
| Elbow/Upper Arm | 40% | 64% | 100% |
| Shoulder | 0% | 9% | 33% |
| Thoracic/Cervical Vertebrae | 0% | 18% | 50% |
| Lumbar/Sacral Vertebrae | 0% | 27% | 33% |
| Jaw | 10% | 18% | 100% |
| At least one site of HO/Overgrowth | 40% | 100% | 100% |

DOI: https://doi.org/10.7554/eLife.40814.004

known to be susceptible in humans (*Table 1*). By P28, all Pdgfrα-R206H mice exhibited HO at multiple sites (*Figure 1G–M*; *Table 1*). HO (*Figure 1I*) and osteochondromas (*Figure 1L*) were often detected at the deltoid tuberosity (also see *Figure 1—figure supplement 1A–C*). Although it remains formally possible that osteochondromas progressed to HO in Pdgfrα-R206H mice, such an occurrence is rare in human FOP (*Deirmengian et al., 2008*; *Kaplan et al., 1993*; *Morales-Piga et al., 2015*). Hence, we speculate that HO of the deltoid tuberosity arises from recruitment of nearby intramuscular or intratendinous R206H-FAPs. Periosteal cells are another possible HO precursor, as Pdgfrα-Cre is also expressed in the periosteum (*Lees-Shepard et al., 2018*). Joint ankylosis was common in P28 Pdgfrα-R206H mice, which weighed an average of 19% less than wild-type littermates (*Figure 1T*; n = 11; p<0.01). At the P42 endpoint of the natural history study, all surviving Pdgfrα-R206H mice were heavily burdened with HO (*Figure 1N–S*) and exhibited ankylosis of multiple joints (*Table 1*). Pdgfrα-R206H mice weighed an average of 35% less than littermate controls at P42 (*Figure 1T*; n = 6; p<0.0001). For humane reasons, Pdgfrα-R206H mice were euthanized if body weight loss from peak weight exceeded 20%. Using both premature death and removal from the study to calculate survival, median survival of Pdgfrα-R206H mice was 39 days, with a significantly reduced survival rate of 43% at P42 (*Figure 1U*; n = 14; p<0.0001). It is likely that reduced survival and substantial weight loss are primarily related to ankylosing HO of the jaw, which was observed in all mice that failed to reach the P42 natural history study endpoint.

## Daily palovarotene treatment ameliorates HO and osteochondroma formation in juvenile Pdgfrα-R206H mice

To assess the effect of daily palovarotene treatment, Pdgfrα-R206H mice were randomized into one of three groups that received daily administration of 0.735 mg/kg palovarotene, 1.47 mg/kg palovarotene, or vehicle alone from P14 through P41. Mean body weight gain in palovarotene-treated Pdgfrα-R206H mice trended lower than vehicle (21% less for the 0.735 mg/kg palovarotene group and 28% less for 1.47 mg/kg palovarotene group) (*Figure 2A*), although this difference was not statistically significant. 47% of vehicle-treated Pdgfrα-R206H mice survived to P42, similar to untreated Pdgfrα-R206H mice from the natural history study (Compare *Figure 2B* to *Figure 1U*). Surprisingly, palovarotene did not improve survival of Pdgfrα-R206H mice, with a P42 survival rate of 27% for 0.735 mg/kg palovarotene-treated mice and 33% for 1.47 mg/kg palovarotene-treated mice (*Figure 2B*). Whereas 100% of Pdgfrα-R206H mice euthanized early or found dead in the vehicle treatment group had HO of the jaw (*Figure 2C*), 9% of 0.735 mg/kg palovarotene-treated mice and 60% of 1.47 mg/kg palovarotene-treated mice euthanized or dead prior to P42 lacked HO of the jaw (*Figure 2C*). In addition, treatment with 1.47 mg/kg palovarotene also reduces survival of wild-type juvenile mice (*Figure 2—figure supplement 1*). These results suggest that palovarotene

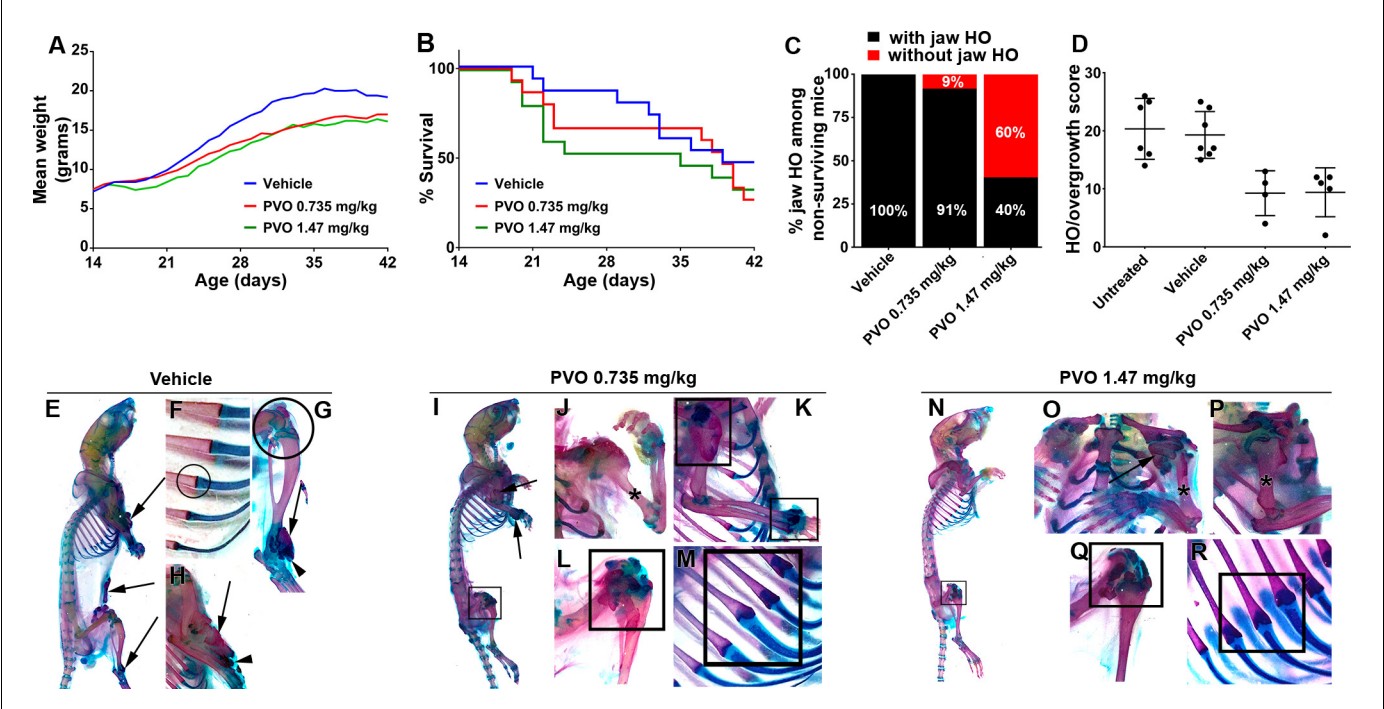

**Figure 2.** Palovarotene reduces the severity of HO in Pdgfrα-R206H mice. (**A**) Body weight of Pdgfrα-R206H mice receiving palovarotene (PVO) trended lower but was not significantly different from those receiving vehicle, as assessed by two-way ANOVA. (**B**) Survival of Pdgfrα-R206H mice receiving PVO was not significantly different that those receiving vehicle, as assessed by log-rank. (**C**) Of Pdgfrα-R206H mice euthanized early or found dead prior to P42, only PVO-treated groups contained mice that lacked jaw HO. (**D**) Comparison of HO/overgrowth burden at P42. Data for untreated mice are from the natural history study. Both PVO-treated groups exhibited a ~ 50% reduction in HO severity. The HO/overgrowth score of each mouse is represented as a single point; group mean (black bar)±standard deviation (error bars) are shown. (**E–R**) Whole mount ABAR skeletal preparations of Pdgfrα-R206H mice at P42. (**E–H**) Vehicle treatment did not alter HO pathogenesis, which presented as bone (arrows) with limited cartilage (arrowheads). (**F**) Rib and (**G**) knee morphology was normal in vehicle-treated Pdgfrα-R206H mice (circles). (**I–M**) 0.735 mg/kg PVO-treated and (**N–R**) 1.47 mg/kg PVO-treated Pdgfrα-R206H mice exhibited reduced HO (arrows), and (**J, O, P**) osteochondromas were often absent (asterisks). (**I, K–N, Q, R**) PVO-treated Pdgfrα-R206H mice commonly presented abnormal shoulder, wrist, knee, and rib morphology at the costochondral junction (boxes).

DOI: https://doi.org/10.7554/eLife.40814.005

The following figure supplement is available for figure 2:

**Figure supplement 1.** Palovarotene reduces survival in juvenile wild-type mice.

DOI: https://doi.org/10.7554/eLife.40814.006

exhibits dose-dependent toxicity in juvenile mice, although the reason for reduced survival is currently unknown.

As extensive fusion of HO to the endogenous skeleton prevented accurate volumetric HO segmentation and quantification, we employed a semi-quantitative scale to score the severity of HO. Using mice stained with Alcian Blue and Alizarin Red, sites of HO and skeletal overgrowth were scored as mild (1), moderate (2), or severe (3) based on estimated total HO volume. A cumulative score per mouse was then calculated by adding the individual score for each site of HO or overgrowth. As observed in the natural history study, vehicle-treated Pdgfrα-R206H mice that survived to the end of the study (P42) exhibited extensive HO at multiple sites (*Figure 2D,E,G,H*; *Table 2*). Total body HO/overgrowth burden was reduced ~50% by palovarotene treatment (*Figure 2D*; *Table 2*), which was not significantly different between the two palovarotene treatment groups. The failure of increased dose to further reduce HO suggests limits to the therapeutic efficacy of palovarotene in FOP mice.

Interestingly, palovarotene exhibited disparate site-specific effects. For example, although palovarotene substantially reduced HO/overgrowth at the elbow/upper arm and ankle (*Figure 2I,J,K,N, O,P*; *Table 2*), palovarotene-treated Pdgfrα-R206H mice exhibited comparable or increased incidence of HO/overgrowth at the knee, shoulder, and spine (*Figure 2I,K,L,N,O,Q*; *Table 2*). The knee

**Table 2.** Incidence of HO/overgrowth in vehicle and PVO-treated P42 Pdgfrα-R206H mice

| Site | Vehicle (n = 7) | 0.735 mg/kg PVO (n = 4) | 1.47 mg/kg PVO (n = 5) |
|---|---|---|---|
| Ankle | 86% | 0% | 20% |
| Knee | 29% | 75% | 80% |
| Hip | 14% | 0% | 40% |
| Wrist | 0% | 50% | 0% |
| Abdomen | 57% | 25% | 40% |
| Forearm | 14% | 0% | 0% |
| Elbow/Upper Arm | 100% | 50% | 0% |
| Shoulder | 57% | 75% | 80% |
| Thoracic/Cervical Vertebrae | 14% | 25% | 80% |
| Lumbar/Sacral Vertebrae | 29% | 75% | 100% |
| Jaw | 43% | 0% | 40% |
| At least one site of HO/Overgrowth | 100% | 100% | 100% |

DOI: https://doi.org/10.7554/eLife.40814.007

joint was most commonly affected in P42 Pdgfrα-R206H mice receiving palovarotene, with three of four mice treated with 0.735 mg/kg palovarotene and four of five mice receiving 1.47 mg/kg palovarotene exhibiting apparent overgrowth at this site (*Figure 2I,L,N,Q*; *Table 2*). In addition, altered rib morphology at the costochondral junction was observed in one of four 0.735 mg/kg palovarotene-treated (*Figure 2M*) Pdgfrα-R206H mice and all five 1.47 mg/kg palovarotene-treated Pdgfrα-R206H mice (*Figure 2R*). Abnormal joint or rib morphology was not observed in vehicle-treated Pdgfrα-R206H mice (*Figure 2F,G*) or untreated Pdgfrα-R206H mice from the natural history study.

## Daily palovarotene treatment disrupts growth plate and synovial joint morphology in juvenile FOP mice

To further characterize the aberrant skeletal growth phenotype of palovarotene-treated Pdgfrα-R206H mice, we conducted μCT and histological analysis of the knee at P42. Whereas overall knee morphology of vehicle-treated Pdgfrα-R206H mice was comparable to wild-type mice (*Figure 3A–D*), three of four 0.735 mg/kg palovarotene-treated Pdgfrα-R206H mice and four of five 1.47 mg/kg palovarotene-treated Pdgfrα-R206H mice exhibited severe osteochondral overgrowth at the knee (*Figure 3E–H*). Histological analyses indicated that the overgrowths emanated from the articular cartilages and associated connective tissues of the distal femur and proximal tibia (*Figure 3F,H*). Similar adverse effects were observed when wild-type mice were administered an identical palovarotene dosing regimen (*Figure 3—figure supplement 1A–F*). Whether palovarotene-induced articular cartilage overgrowth is related to the pro-proliferative effects of retinoids on chondrocytes in other experimental settings (*Enomoto et al., 1990*; *Kwon et al., 2011*) requires further investigation. Regardless of mechanism, these data indicate that palovarotene exhibits skeletal toxicity that is consistent with known effects of chronic retinoid treatment on synovial joint tissue in animal models (*Gartner et al., 1970*; *Clark and Seawright, 1968*; *Kubo et al., 2002*) and in humans (*Noyes et al., 2016*; *Vahlquist, 1992*).

As retinoids are known regulators of growth plate chondrogenesis (*De Luca et al., 2000*; *Williams et al., 2010*; *Uezumi et al., 2009*), we next assessed the effect of daily palovarotene treatment on tibial growth plate morphology. Histological evaluation of the proximal tibia showed that vehicle-treated Pdgfrα-R206H mice exhibited growth plate, articular cartilage, and trabecular bone morphology that was comparable to wild-type mice (*Figure 4A–F*). In contrast, three of four Pdgfrα-R206H mice treated with 0.735 mg/kg palovarotene and four of five Pdgfrα-R206H mice treated with 1.47 mg/kg palovarotene lacked a proximal tibia growth plate and exhibited thickened articular cartilage (*Figure 4G–L*). Although trabeculation also was reduced in palovarotene-treated mice that lacked a growth plate (*Figure 4G–L*), it is unclear if reduced trabeculation is a direct effect of palovarotene or resulted from reduced load-bearing associated with altered knee morphology. Growth

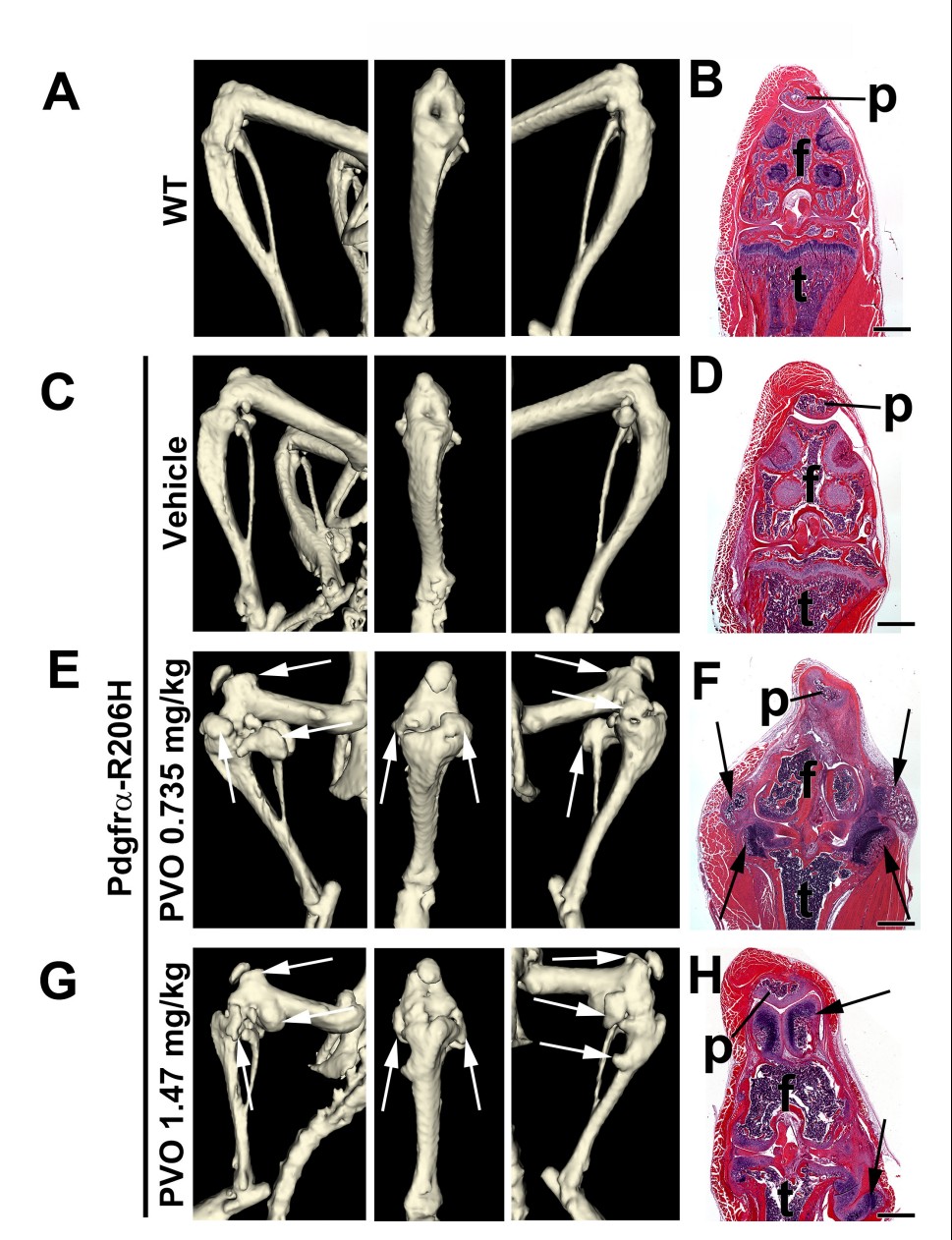

**Figure 3.** Palovarotene adversely affects synovial joints in juvenile Pdgfrα-R206H mice. (A, C, E, G) μCT images of P42 mouse knees displaying lateral, frontal, and medial orientation. (B, D, F, H) Hematoxylin and eosin (H and E) staining of the same knee (p, patella; f, femur, t, tibia), oriented on the frontal plane. (A, B) WT knee morphology is comparable to that of (C, D) vehicle-treated Pdgfrα-R206H mice. (E–H) PVO-treated Pdgfrα-R206H mice exhibited overgrowth of knee articular cartilage, as detected by μCT (white arrows) and histology (black arrows). Scale bars = 1 mm for (B, D, F, H).

DOI: https://doi.org/10.7554/eLife.40814.008

The following figure supplement is available for figure 3:

**Figure supplement 1.** Palovarotene treatment results in skeletal toxicity and reduced survival in wild-type mice.

DOI: https://doi.org/10.7554/eLife.40814.009

plate loss was also observed in palovarotene-treated wild-type mice (*Figure 3—figure supplement 1A–F*). Measurement of the tibial growth plate zones of Pdgfrα-R206H mice revealed normal proliferative zone width but reduced hypertrophic zone width (*Figure 4M* p=0.01), as was reported for *Acvr1[R206H]FlEx/+*;Prrx1-Cre mice (*Chakkalakal et al., 2016*). However, whether palovarotene normalized hypertrophic zone length in Pdgfrα-R206H mice, as was observed in *Acvr1[R206H]FlEx/+*;Prrx1-Cre

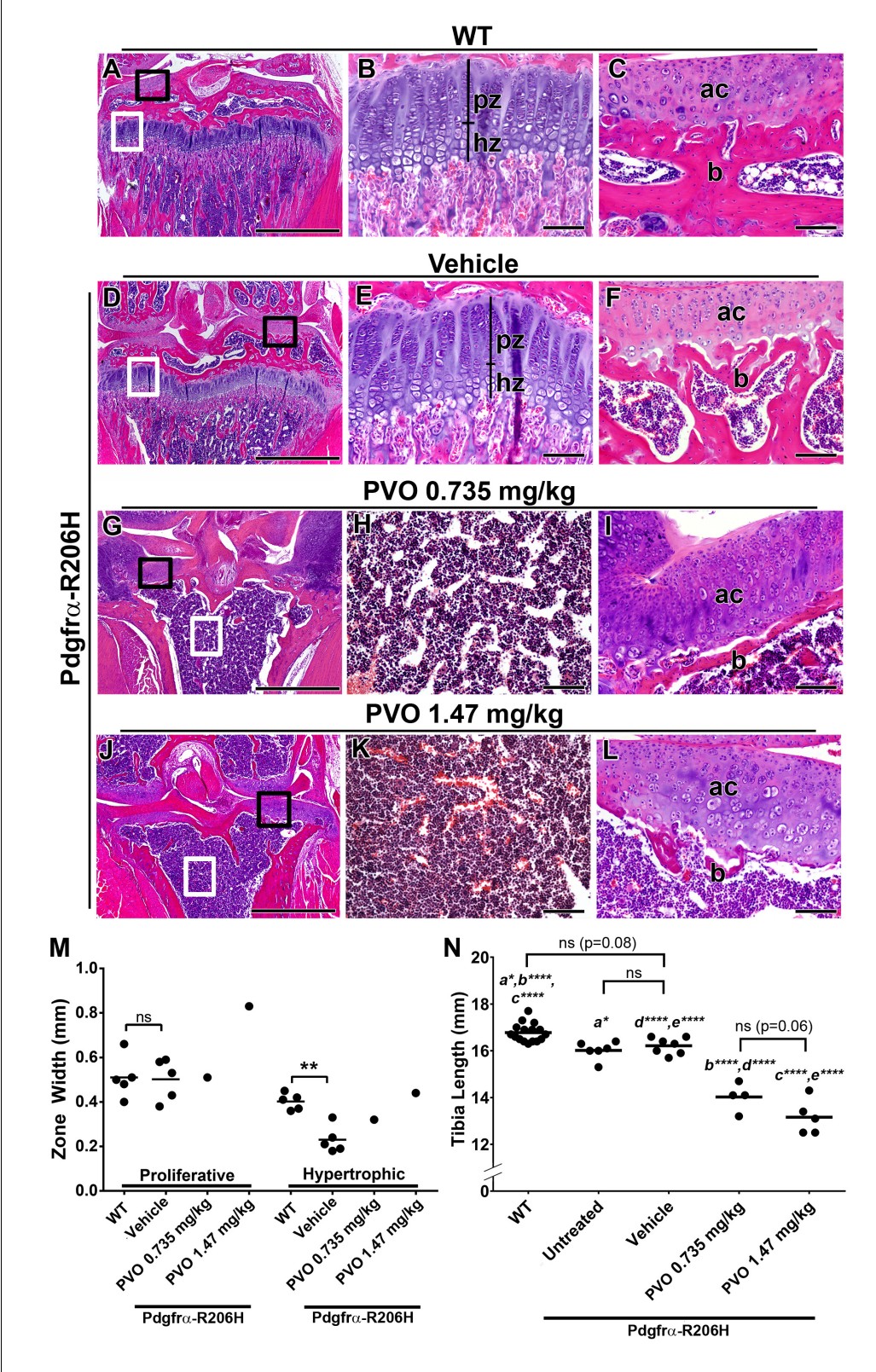

**Figure 4.** Palovarotene results in growth plate loss in juvenile Pdgfrα-R206H mice. (**A–L**) H and E analysis of the proximal P42 tibia, oriented on the frontal plane. (**A, D**) Low magnification images with the growth plate (white boxes) and articular cartilage (black boxes) denoted. Corresponding high magnification images of the (**B, E**) growth plate and (**C, F**) articular cartilage revealed that WT mice and vehicle-treated Pdgfrα-R206H mice exhibit similar growth plate (pz = proliferative zone; hz = hypertrophic zone), subchondral bone (**b**), and articular cartilage (ac) morphology. (**G, J**) Low

*Figure 4 continued on next page*

*Figure 4 continued*

magnification images showing growth plate loss (white boxes) and thickened articular cartilage (black boxes) in PVO-treated Pdgfrα-R206H mice. Corresponding high magnification images revealed (H, K) an open marrow space lacking trabeculation, and (I, L) thinner subchondral bone (b) coupled with thickened articular cartilage (ac). (M) Growth plate proliferative and hypertrophic zone width at P42. (N) Tibial length of WT and Pdgfrα-R206H mice at P42. Each dot represents data from a single mouse, and group mean is represented by a horizontal bar. Statistical significance was determined by one-way ANOVA (ns, not significant; *=p ≤ 0.05; **=p ≤ 0.01; ***=p ≤ 0.001; ****=p ≤ 0.0001). Scale bars = 1 mm for A, D, G, J and 100 μm for B, C, E, F, H, I, K, L.

DOI: https://doi.org/10.7554/eLife.40814.010

The following figure supplements are available for figure 4:

**Figure supplement 1.** Growth plate trauma in palovarotene-treated Pdgfrα-R206H mice.

DOI: https://doi.org/10.7554/eLife.40814.011

**Figure supplement 2.** Palovarotene reduced skull length in Pdgfrα-R206H mice.

DOI: https://doi.org/10.7554/eLife.40814.012

mice (*Chakkalakal et al., 2016*), could not be assessed as only one mouse from each palovarotene treatment group retained a growth plate at P42 (*Figure 4M*). Interestingly, the single mouse from each palovarotene treatment group that retained a growth plate at P42 exhibited normal articular cartilage and trabecular bone morphology (*Figure 4—figure supplement 1A–D*). However, boney bridges were observed within the growth plates of these mice *Figure 4—figure supplement 1B,D*), which are suggestive of prior growth plate trauma (*Wattenbarger et al., 2002*; *Xian et al., 2004*). Given that alteration of cartilage matrix composition is a known effect of retinoids (*Campbell and Handley, 1987*; *Von den Hoff et al., 1993*; *Ostendorf et al., 1995*), we speculate that these two mice represent an intermediate stage prior to the complete growth plate loss observed in all other palovarotene-treated Pdgfrα-R206H mice.

Pdgfrα-R206H mice exhibited a 5% reduction in tibial length, as compared to wild-type mice (*Figure 4N*; p=0.02). In comparison to vehicle, which did not affect tibial length in Pdgfrα-R206H mice, tibial length was shortened by 13% (p<0.001) in response to 0.735 mg/kg palovarotene and 19% (p<0.001) in response to 1.47 mg/kg palovarotene (*Figure 4N*). Although the trend indicated a dose-dependent decrease in tibial length, differences between doses did not reach statistical significance (p=0.06). While differences in total body length and spine length between treatment groups at the end of the study were not statistically significant, these parameters trended lower in palovarotene-treated Pdgfrα-R206H mice, compared to vehicle-treated mice (*Figure 4—figure supplement 2A,B*). In contrast, palovarotene-treated Pdgfrα-R206H mice exhibited stouter snouts compared to wild-type and vehicle-treated Pdgfrα-R206H mice, likely due to an effect of palovarotene on nasal cartilage (*Figure 4—figure supplement 2C,D*).

## Palovarotene inhibits activin A-induced chondrogenic and osteogenic differentiation of R206H-FAPs but does not permanently alter FAP behavior

To examine the ability of palovarotene to reduce the skeletogenic capacity of R206H-FAPs, we induced chondrogenic differentiation by micromass culture of FACS-isolated and expanded R206H-FAPs. Inclusion of ActA-mAb in the culture media resulted in loss of Alcian Blue staining, demonstrating that chondrogenic differentiation of R206H-FAPs, like osteogenic differentiation (*Lees-Shepard et al., 2018*), is dependent on serum activin A (*Figure 5A*). Further, the chondrogenic capacity of R206H-FAPs was greatly enhanced by inclusion of 25 ng/mL activin A (~1 nM of $\beta_A\beta_A$ dimers; *Figure 5B*). Compared to media only, palovarotene treatment resulted in a dose-dependent inhibition of chondrogenic differentiation (*Figure 5A*). Notably, palovarotene also effectively inhibited the ability of exogenous activin A to induce chondrogenic differentiation of R206H-FAPs, with 10 nM palovarotene displaying similar efficacy as treatment with ActA-mAb (*Figure 5A,B*). We next sought to determine whether treatment with palovarotene inhibits R206H-FAP-driven HO using an established transplantation paradigm (*Lees-Shepard et al., 2018*). SCID hosts receiving daily dosing of 1.47 mg/kg palovarotene exhibited a ~ 70% reduction in HO volume following transplantation of R206H-FAPs, as compared to untreated or vehicle-treated SCID hosts (*Figure 5C,D,F*)

Using both cell culture and transplantation assays, previous studies have shown that a 2- to 3 day pretreatment of bone marrow-derived mesenchymal stem cells with the RARγ agonists CD1530

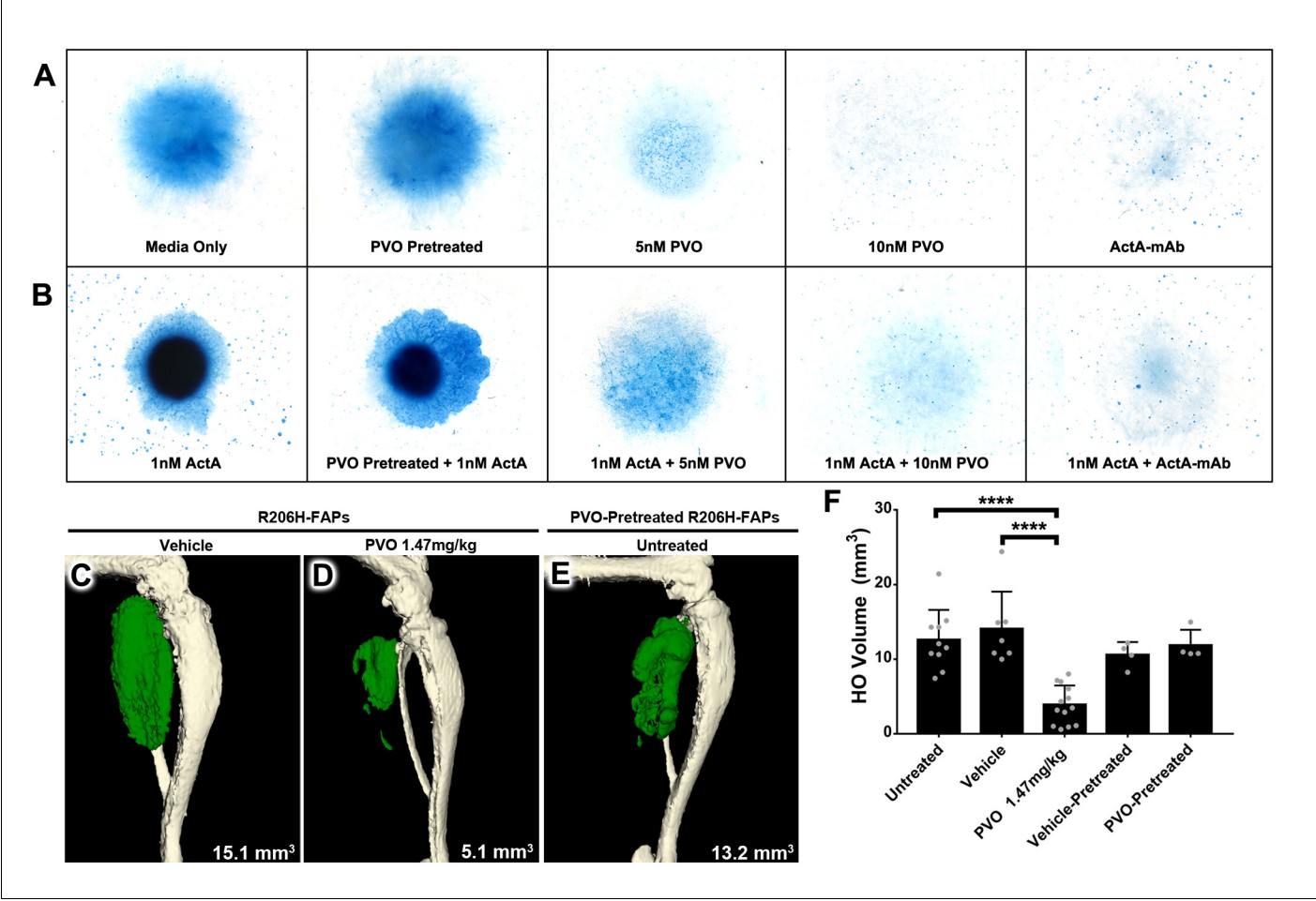

**Figure 5.** PVO treatment inhibits chondrogenic and osteogenic differentiation of R206H-FAPs. (**A, B**) Chondrogenic differentiation of high-density R206H-FAPs cultures in (**A**) base media or (**B**) media with 25 ng/mL (~1 nM) activin A was assessed by Alcian Blue staining to detect cartilage matrix proteoglycans at day 10 (n = 3 experiments). ActA-mAb was used at ~7 fold molar excess to activin A ligand (inhibin $\beta_A$ dimers). (**C–E**) µCT of representative distal hindlimbs 14 days after transplantation of R206H-FAPs into SCID hosts. HO is pseudocolored green and volume given in $mm^3$. (**F**) Quantification of HO volume at day 14 post-transplantation of R206H-FAPs. Each dot represents a single transplantation with group mean (black bar) and ± standard deviation (error bars) shown. Statistical significance was assessed by one-way ANOVA; ****=$p \leq 0.0001$. PVO-pretreated FAPs received 1 µM PVO during the 7 day expansion period prior to in vitro or transplantation assays.

DOI: https://doi.org/10.7554/eLife.40814.013

(*Shimono et al., 2011*) or NRX204647 (*Shimono et al., 2014*) prevents their subsequent differentiation into cartilage and bone, even in the presence of supraphysiological concentrations of osteogenic BMPs (*Shimono et al., 2014*; *Shimono et al., 2011*). To determine whether pretreatment of R206H-FAPs with palovarotene inhibits skeletogenic differentiation, we treated FACS-isolated R206H-FAPs with vehicle or 1 µM palovarotene during the standard, 7 day, expansion period. Following pretreatment, R206H-FAPs were subjected to in vitro micromass culture or transplanted into untreated SCID hosts. Palovarotene-pretreated R206H-FAPs displayed no overt reduction in chondrogenic differentiation (*Figure 5A,B*) and no statistically significant difference in HO forming capacity (*Figure 5E,F*).

## Reduced expansion of transplanted R206H-FAPs is associated with a reduction in HO

Our previous data were consistent with the notion that the density or quantity of intramuscular R206H-FAPs is correlated with the severity of injury-induced HO and the age-of-onset of spontaneous HO (*Lees-Shepard et al., 2018*). Here, we used cell transplantation and bioluminescence

imaging to directly establish the relationship between R206H-FAP cell number and resulting HO. Cell titration experiments determined that under the conditions employed, skeletogenic differentiation of R206H-FAPs requires transplantation of >20,000 cells, and above this threshold number, mean HO volume was proportional to the number of R206H-FAPs transplanted (*Figure 6—figure supplement 1*). Given that palovarotene (*Chakkalakal et al., 2016*; *Wheatley et al., 2018*) and activin A (*Namwanje and Brown, 2016*) can affect cell proliferation and survival in other contexts, we first characterized population growth dynamics of wild-type and R206H-FAPs following transplantation, and subsequently tested whether palovarotene treatment and sequestration of activin A influenced population growth of R206H-FAPs.

Longitudinal live animal imaging revealed that the number of wild-type FAPs almost doubled between days 1 and 3 and then declined by 73% between days 3 and 5, with only 9% of the peak population surviving to day 7 (*Figure 6A,E*). By contrast, R206H-FAPs exhibited enhanced survival at day 1 and more rapid expansion from days 1 to 3. Notably, R206H-FAPs continued to expand from days 3 to 5 (*Figure 6B,E*). The R206H-FAP population declined between days 5 and 7, coinciding with the onset of bone formation (*Figure 6B,E*). This decline is likely attributable to the death of chondrocytes and pre-osteoblasts that accompanies normal endochondral bone formation and to a reduction in detectable luminescence from FAP-derived osteocytes as they become encased in bone matrix.

R206H-FAPs transplanted into ActA-mAb-treated SCID hosts did not form HO and exhibited population dynamics that were not significantly different from that of wild-type FAPs (*Figure 6C,E*; *Figure 6—source data 1*). This agrees with our previous endpoint histology showing that, in the absence of activin A, transplanted R206H-FAPs assumed a wild-type-like interstitial location adjacent to regenerated myofibers (*Lees-Shepard et al., 2018*). Palovarotene treatment did not significantly reduce survival of R206H-FAPs at day 1 after transplantation but did significantly limit their subsequent expansion. However, R206H-FAP numbers remained significantly higher in palovarotene-treated hosts than in untreated or ActA-mAb-treated hosts (*Figure 6D,E*; *Figure 6—source data 1*). For both untreated and palovarotene-treated mice, peak FAP number (typically occurring between days 3 and 5) strongly correlated with severity of HO formation at day 10 (*Figure 6F*; $R^2 = 0.85$ for untreated and 0.83 for PVO treated groups). Thus, it appears that the efficacy of palovarotene in the transplantation paradigm is at least partially attributable to limiting pathogenic expansion and survival of R206H-FAPs.

## Discussion

The longitudinal natural history analysis presented herein revealed the progressive nature of FOP pathogenesis in Pdgfrα-R206H mice. In addition to defining the time course of HO progression through 6-weeks-of-age and the effects of progressive HO on survival and morbidity, the analysis revealed the formation of osteochondromas in Pdgfrα-R206H mice, a phenotype present in FOP patients but not previously described in other FOP mouse models (*Chakkalakal et al., 2016*; *Dey et al., 2016*; *Hatsell et al., 2015*; *Lees-Shepard et al., 2018*). It is also important to note that targeting expression of *Acvr1^R206H* to FAPs did not result in regional or tissue restricted HO, as occurs when *Acvr1^R206H* is expressed by Prrx1+, Mx1+, or Scx+ cell populations (*Chakkalakal et al., 2016*; *Dey et al., 2016*). Hence, Pdgfrα-R206H mice represent a faithful mouse model of juvenile FOP that is well-suited for preclinical drug testing.

In two important respects, results described here differ from previous findings of palovarotene effects on juvenile *Acvr1^[R206H]FlEx/+*;Prrx1-Cre mice (*Chakkalakal et al., 2016*). First, palovarotene treatment of juvenile Pdgfrα-R206H mice resulted in severe skeletal toxicity, including growth plate loss and synovial joint overgrowth. In contrast, whereas both *Chakkalakal et al., 2016* and the present study documented toxic effects of palovarotene on the skeleton of wild-type mice, palovarotene treatment actually ameliorated skeletal growth deficits in *Acvr1^[R206H]FlEx/+*;Prrx1-Cre mice. Second, while palovarotene reduced severity of whole-body HO and inhibited osteochondroma formation in Pdgfrα-R206H mice, palovarotene treatment of *Acvr1^[R206H]FlEx/+*;Prrx1-Cre mice resulted in more marked reductions in HO. Phenotypic differences in palovarotene effects on these distinct FOP mouse models are likely related to study-specific dosing schedules, the developmental stage at which treatment was initiated, and the endpoints at which analyses were conducted. *Chakkalakal et al., 2016* administered palovarotene by oral gavage, first, by daily dosing of

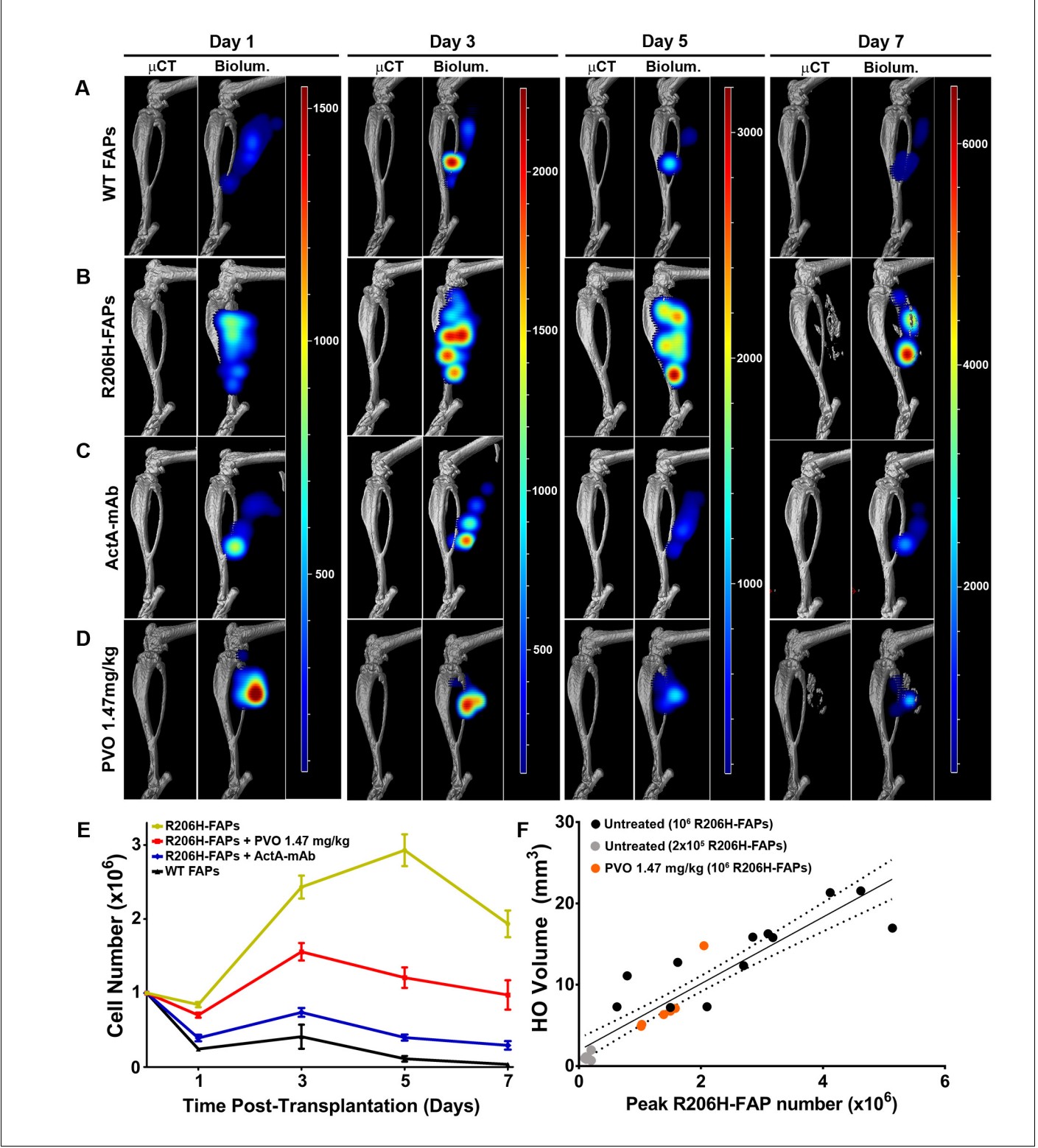

**Figure 6.** Activin sequestration restores wild-type population dynamics to bioluminescent R206H-FAPs. (**A–D**) 3D tomographic bioluminescent source reconstruction following transplantation of $10^6$ bioluminescent FAPs into SCID hosts. Paired images show μCT alone (left panel) and μCT combined with the corresponding 3D bioluminescent reconstruction (right panel). For each experimental group, the same mouse is shown from days 1–7. (**E**) Graphical representation of bioluminescent FAP population dynamics following transplantation into SCID hosts. Error bars represent ±standard error of the mean; see *Figure 6—source data 1* for statistical analysis. (WT, n = 5; R206H-FAPs, n = 12; R206H-FAPs + PVO 1.47 mg/kg, n = 6; R206H-

*Figure 6 continued on next page*

*Figure 6 continued*

FAPs + ActA mAb, n = 4). (**F**) Linear regression analysis of the relationship between HO volume and peak FAP number. Untreated SCID hosts were transplanted with either $10^6$ (black dots; n = 12) or $2 \times 10^5$ (grey dots; n = 6) R206H-FAPs, and PVO-treated SCID hosts were transplanted with $10^6$ (red dots; n = 6) R206H-FAPs. Line of best fit (black line) and 95% confidence interval (dotted lines) are shown. $R^2$ = 0.85 for untreated and 0.83 for PVO 1.47 mg/kg-treated groups.

DOI: https://doi.org/10.7554/eLife.40814.014

The following source data and figure supplements are available for figure 6:

**Source data 1.** Two-way ANOVA analysis of bioluminescent FAP population dynamics *eLife's* transparent reporting form.

DOI: https://doi.org/10.7554/eLife.40814.017

**Figure supplement 1.** HO induction requires a threshold number of R206H-FAPs.

DOI: https://doi.org/10.7554/eLife.40814.015

**Figure supplement 2.** Quantification of FAP bioluminescence.

DOI: https://doi.org/10.7554/eLife.40814.016

lactating females at a dose of 50 µg/mouse from the day of delivery to P15, and subsequently, by alternate-day dosing of *Acvr1*[R206H]FlEx/+;Prrx1-Cre pups with 20 µg/mouse palovarotene from P16 to P30. The present study utilized bodyweight-adjusted daily dosing from P14 to P42 by intraperitoneal injection of 0.735 mg/kg (equivalent to a starting and ending average dose of ~5 µg and ~11 µg per mouse, respectively) and 1.47 mg/kg (~10 to~22 µg/mouse). Importantly, palovarotene-induced skeletal toxicity is neither unique to the Pdgfrα-R206H mouse model nor to intraperitoneal administration, as apparently similar skeletal overgrowths occurred, but were not noted, in an independent study by *Inubushi et al., 2018*, in which mice were administered comparable daily doses of palovarotene via oral gavage from 2- to 6-weeks-of-age (*Inubushi et al., 2018*), as in the present study. Since the palovarotene doses used in the current study and by *Inubushi et al., 2018* were less than, or comparable to, that used by *Chakkalakal et al., 2016* on a per-dose basis, it is reasonable to suggest that severe skeletal toxicity was related to greater overall palovarotene exposure associated with daily dosing. The overall length of the treatment window and developmental stage of exposure also appear to be critical variables, although distinguishing the relative contribution of each parameter is difficult. Thus, *Inubushi et al., 2018* showed that reducing the window of palovarotene administration to 3- to 6-weeks-of-age did not result in reductions in tibia and femur length, which are characteristic of the 2- to 6 week exposure period, or in the loss of the distal femur growth plate (*Inubushi et al., 2018*). The developmental window of palovarotene exposure might also explain the comparatively lower efficacy in reducing HO reported here, despite the daily dosing regimen, as a palovarotene dosing schedule that eliminated exposure from P0 to P14 was less effective at inhibiting HO (*Chakkalakal et al., 2016*). Collectively, these data indicate that the developmental stage and duration of exposure to palovarotene, as well as dosing interval, all contribute to the extent of palovarotene efficacy and skeletal toxicity.

Pretreatment of bone marrow-derived mesenchymal stem cells with the RARγ agonists CD1530 or NRX204647 confers protection from BMP2-induced HO (*Shimono et al., 2014*; *Shimono et al., 2011*). If this were generally true for RARγ agonists, the risk of palovarotene-induced skeletal toxicity would likely be mitigated by the requirement for only short-term administration. However, it is clear from the present data that pretreatment of R206H-FAPs with palovarotene for 7 days does not render them refractory to skeletogenic differentiation. It remains unclear whether these disparate experimental outcomes are attributable to differences between palovarotene and other RARγ agonists, FAPs and bone marrow-derived mesenchymal stem cells, or the mechanism through which BMP2 and activin A induce HO.

Antibody-mediated activin inhibition is also being evaluated for safety and efficacy in a clinical trial for FOP. The data presented herein represents the first direct comparison of the ability of these two therapeutic modalities to modulate the behavior of FAPs, a major contributor to heterotopic skeletogenesis. By live-animal imaging we showed that treatment of SCID hosts with a blocking antibody to activin A restored wild-type population dynamics to transplanted R206H-FAPs, whereas palovarotene treatment resulted in a more modest, but statistically significant, reduction in the kinetics and extent of R206H-FAP population expansion. Consistent with this latter finding, recent studies showed that palovarotene reduces proliferation of lesional cells in injured FOP mice (*Chakkalakal et al., 2016*) and in a rat model of blast injury and amputation-induced HO

(*Pavey et al., 2016*). Although the present study did not distinguish between effects on cell proliferation and survival, it is reasonable to predict that inhibition of FAP proliferation by palovarotene represents a common mechanism of action in the present and previous studies. Our finding that total HO volume was tightly correlated with peak R206H-FAP number following transplantation predicts that therapeutic approaches that limit R206H-FAP proliferation, in addition to targeting signaling events associated with endochondral bone formation, would show greatest efficacy. More generally, targeting of FAP proliferation might have implications beyond FOP, as dysregulation of FAP population dynamics has been implicated in distinct pathological conditions. Specifically, FAP numbers rapidly increase following muscle injury, and the subsequent failure to appropriately reduce their numbers after initial expansion is associated with FAP-mediated fibrosis of muscle (*Joe et al., 2010*; *Lemos et al., 2015*; *Mueller et al., 2016*; *Uezumi et al., 2011*; *Uezumi et al., 2010*). Thus, as inappropriate FAP survival and proliferation appear to represent a common indicator of pathogenic versus pro-regenerative FAP function (*Ito et al., 2013*; *Lemos et al., 2015*; *Fiore et al., 2016*), and FAPs are widely distributed in non-muscle tissues and organs (*Wosczyna et al., 2012*), palovarotene and activin inhibition might also show efficacy in diseases or conditions characterized by fibrosis of muscles or other tissues.

## Materials and methods

### Mouse crosses and genotyping

All animal procedures were reviewed and approved by the University of Connecticut Institutional Animal Care and Use Committee. Tie2-Cre transgenic mice (*Kisanuki et al., 2001*) were a gift of Dr. Tom Sato (UT Southwestern). Generation of $Acvr1^{tnR206H}$ (*Lees-Shepard et al., 2018*) and $R26^{NG}$ (*Yamamoto et al., 2009*) knockin mice has been previously reported. Pdgfrα-Cre transgenic mice (*Roesch et al., 2008*) (Tg(Pdgfra-cre)1Clc) and $R26^{luc}$ (*Safran et al., 2003*) mice (FVB.129S6(B6)-*Gt (ROSA)26Sor^{tm1(Luc)Kael}*/J) were obtained from Jackson Laboratories. The SCID Hairless Outbred mice (SHO-*Prkdc^{scid}Hr^{hr}*) used for transplantation were obtained from Charles River.

$Acvr1^{tnR206H/+}$;$R26^{NG/+}$;Pdgfrα-Cre mice were generated as previously described (*Lees-Shepard et al., 2018*), and $Acvr1^{tnR206H/+}$;$R26^{luc/+}$;Tie2-Cre and $R26^{luc/+}$;Tie2-Cre mice were generated using the same breeding scheme. Adult mice between 8- and 16-weeks-of-age were used for all experiments, and male and female mice were used interchangeably in all studies.

Mice were genotyped by PCR and reporter fluorescence, as previously described (*Lees-Shepard et al., 2018*). The following primers were used for genotyping $R26^{luc}$ mice: 5′-CGGTATCG TAGAGTCGAGGCC-3′ and 5′-CAGGGCGTATCTCTTCATAGCC-3′. Mouse colonies were managed using SoftMouse Colony Management software (softmouse.net).

### Reagents

Activin A was obtained from R and D Systems (Minneapolis, MN). A monoclonal antibody against activin A (ActA-mAb), which was provided by Acceleron Pharma (Cambridge, MA), was described previously (*Lees-Shepard et al., 2018*). Palovarotene was either provided, preformulated, by Clementia Pharmaceuticals (Montreal, Canada) or obtained from MedChemExpress (Monmouth Junction, NJ). Palovarotene dosing formulations were prepared as follows: 0.0735 mg/mL palovarotene in 0.735% DMSO with 4% Tween 80 in PBS (pH 7.4); 0.147 mg/mL palovarotene in 1.47% DMSO with 4% Tween 80 in PBS (pH 7.4). Vehicle dosing formulation was 1.47% DMSO with 4% Tween 80 in PBS (pH 7.4). The palovarotene dosing formulations prepared by Clementia were shipped on dry-ice, pre-aliquoted for daily use, and dosing formulations using palovarotene obtained from MedChemExpress were prepared in-house. All mice receiving palovarotene or vehicle were administered 10 mL per kg bodyweight.

### Daily palovarotene dosing

$Acvr1^{tnR206H/+}$;$R26^{NG/+}$;Pdgfrα-Cre mice were randomized into three study groups that received daily intraperitoneal (IP) injections of either 0.735 mg/kg palovarotene, 1.47 mg/kg palovarotene, or vehicle alone from P14 to P41. Palovarotene doses were determined by Clementia Pharmaceuticals and correspond to approximate adult human equivalent doses of 3.6 mg and 7.2 mg (*Nair and Jacob, 2016*). Similar doses were effective at inhibiting osteochondroma formation in a mouse

model of hereditary exostosis (*Inubushi et al., 2018*). IP injection was selected as the route of administration to avoid the possibility that physical manipulations associated with oral gavage would induce HO of the jaw. Untreated control littermates lacking either *Acvr1^{tnR206H}* or Pdgfrα-Cre were included for comparison (these mice were phenotypically wild-type and are referred to as wild-type throughout). Investigators were blinded as to the identity of the groups until completion of the analysis. Mice were removed from the study and euthanized if body weight loss from peak weight exceeded 20%.

### μCT imaging analyses and measurements

μCT images were taken with an IVIS Spectrum-CT (Model 128201; Perkin Elmer, Hopkinton, MA) using either the medium resolution (75 μm voxel size; estimated radiation dose of 132 mGy; 210 s scan time) or standard resolution (150 μm voxel size; estimated radiation dose of 52.8 mGy; 140 s scan time) setting while mice were under isoflurane anesthesia. μCT images were generated and analyzed using Living Image 4.5 software (Perkin Elmer, Hopkinton, MA) or 3D Slicer software (http://www.slicer.org). 3D Slicer was used to measure body, spine, skull, and tibial length. Body length was measured from the anterior tip of the nasal bone to the base of sacral vertebrae 4. Skull length was measured from the anterior tip of the nasal bone to the most posterior point of the cranium, and spine length was measured from the foramen magnum to the base of sacral vertebrae 4. HO volume was quantified as previously described (*Lees-Shepard et al., 2018*).

### Histology

Tissues were fixed immediately post-dissection in either 4% paraformaldehyde or 10% neutral buffered formalin (Sigma, St. Louis, MO). Samples were then decalcified in 12% EDTA (pH 7.2) and processed for paraffin embedding. Histological analysis was performed on 10 μm sections. Deparaffinized sections were stained with hematoxylin and eosin using standard methods. Histological specimens were imaged on a Nikon E600 microscope (Nikon, Tokyo, Japan) equipped with a Spot RT3 camera and Spot Advanced image capture software (Diagnostic Instruments, Sterling Heights, MI). Image processing and assembly were performed using Photoshop (Adobe, San Jose, CA).

### Whole-mount skeletal preparations and HO scoring

Eviscerated adult mice were processed for whole-mount Alcian Blue and Alizarin Red skeletal staining as previously described (*Lees-Shepard et al., 2018*). Skeletal overgrowth and boney/cartilaginous HO lesions were assessed by visual inspection and assigned a score of 1 (small), 2 (intermediate) or 3 (large) based on estimated total volume. Each mouse was independently scored by a minimum of three researchers, blinded as to the identity of the groups, and the average score was recorded. Total HO/overgrowth load was calculated by adding the cumulative score of HO and skeletal overgrowth at all sites per mouse. Whole-mount Alcian Blue and Alizarin Red skeletal samples were imaged on a Pentax K-30 camera (Ricoh Imaging, Tokyo, Japan). Image processing and assembly was performed using Photoshop.

### Fluorescence activated cell sorting (FACS) and FAP expansion

Details of skeletal muscle dissection and isolation of FAPs have been previously described (*Biswas and Goldhamer, 2016*; *Lees-Shepard et al., 2018*; *Wosczyna et al., 2012*). Sorting was performed on a FACS Aria II (BD Biosciences, Franklin Lakes, NJ) equipped with 407, 488, and 633 lasers. FACS-isolated FAPs were seeded at a density of 2000 cells/cm$^2$ onto tissue culture flasks (Nunc, Rochester, NY) in Dulbecco's Modified Eagle Medium (DMEM; Life Technologies, Carlsbad, CA) with 50 U/mL Penicillin and 50 μg/mL Streptomycin (Pen/Strep; Gibco, Billings, MT) and 20% HyClone fetal bovine serum (FBS), characterized (GE Healthcare, Chicago, IL; Lot# A00168). FAPs were maintained at 37°C in a humidified atmosphere at 5% $CO_2$. Media was changed every other day. All experiments utilized FAPs passaged fewer than three times.

### Chondrogenic assay

FACS-isolated and expanded FAPS were resuspended at $2 \times 10^7$ cells/mL and plated in 10 μL high-density micromass dots onto 35 mm tissue culture dishes (Nunc, Rochester, NY). Following attachment, DMEM/F12 media (Life Technologies, Carlsbad, CA) containing 5% FBS plus Pen/Strep was

added. Micromass cultures were fed every other day as described for each experiment. After 10 days of culture, cells were fixed with 10% neutral buffered formalin (Sigma, St. Louis, MO) and stained with Alcian Blue, as previously described (*Gay and Kosher, 1984*), to detect cartilage-specific proteoglycans.

### Transplantation and drug treatment

FACS-isolated and expanded FAPs were resuspended in 50 μL of ice-cold 1X Dulbecco's Phosphate-Buffered Saline (DPBS; Gibco, Billings, MT) and injected into the gastrocnemius muscle of SCID mice as previously described (*Lees-Shepard et al., 2018*). In most cases, both gastrocnemius muscles of an individual mouse were injected. For a given treatment group, variation in HO volume within and between mice was not statistically different, and each injection was treated as an independent event for statistical analysis. The gastrocnemius muscle was pinch-injured the day of transplantation using 3500 – 3700 grams of force applied with a Randall Selitto Paw Pressure Test Apparatus (IITC Life Science, Woodland Hills, CA). Treated SCID host mice received a single subcutaneous dose of ActA-mAb (10 mg/kg) on the day of injury, or daily IP injections of 1.47 mg/kg palovarotene or vehicle, beginning 3 days prior to injury and transplantation.

### In vivo bioluminescence imaging

Bioluminescence images were acquired using an IVIS Spectrum-CT and analyzed with Living Image 4.5 software. SCID hosts were injected IP with D-luciferin (Perkin Elmer, Hopkinton, MA) at 150 mg/kg prior to bioluminescence imaging. The bioluminescent light emission plateau was empirically determined to be 12 –16 min after D-luciferin substrate injection. Animals were anesthetized using the built-in XGI-8 Gas Anesthesia System with oxygen containing 2% isoflurane and placed into the imaging chamber. For 3D Diffuse Luminescent Imaging Tomography (DLIT) reconstruction, a μCT image was obtained first using the medium resolution setting (75 μm voxel size; estimated radiation dose 132 mGy; 210 s scan time) for subsequent surface tomography reconstruction. The imaging parameters are comprised of a series of 2D bioluminescence surface radiance images at 560, 580, 600, 620 and 640 nm with a field of view of 6.5 cm. The DLIT reconstruction algorithm in Living Image 4.5 software utilizes the acquired μCT image to establish the air/tissue boundary upon which it superimposes a 3D reconstruction of the bioluminescent signal, which is based on differential surface bioluminescence signal intensity across the firefly luciferase emission spectrum. The DLIT reconstruction was normalized to tissue absorption spectra and calibrated to estimate cell number based on source depth, signal intensity, and light emission per cell (see well-plate quantification below).

### Quantification of FAP bioluminescence in vitro

To estimate luminescence output per cell for DLIT reconstruction, serial dilutions of the luciferase-expressing FAP population to be transplanted were plated into black-walled 24-well plates. For bioluminescence quantification, media was replaced with DMEM containing 0.3 mg/ml D-luciferin (Perkin Elmer, Hopkinton, MA). The luminescence signal was measured using an IVIS Spectrum-CT. Serial images were acquired with an open emission filter at 1 min intervals to determine peak signal. Light emission (photons/second) was quantified and normalized as a function of cell number using the Well Plate Quantification function of Living Image 4.5 software. A representative example is shown in *Figure 6—figure supplement 2*.

### Statistical analysis

Statistical analysis was performed using GraphPad Prism (GraphPad, La Jolla, CA). All numerical values are presented as mean values ± the standard error of the mean or standard deviation, and one or two-way ANOVA was used to determine significance, as described in the corresponding figure legends. Survival was assessed by log-rank. Differences were considered significant at $p < 0.05$.

## Acknowledgements

We thank members of the Goldhamer lab for helpful comments throughout the course of this work, and Dr. Isabelle Lemire (Clementia Pharmaceuticals) for critical input on study design and careful critique of the manuscript. We also thank FACS facility scientist, Dr. Wu He, for expert technical

assistance, and Acceleron Pharma for the anti-activin A monoclonal antibody. This work was supported by a sponsored research agreement with Clementia Pharmaceuticals and a grant from the NIH to DJG (R01AR057371).

## Additional information

### Competing interests

David J Goldhamer: The work was funded, in part, by a sponsored research agreement with Clementia Pharmaceuticals. The other authors declare that no competing interests exist.

### Funding

| Funder | Grant reference number | Author |
| --- | --- | --- |
| National Institute of Arthritis and Musculoskeletal and Skin Diseases | R01AR057371 | David J Goldhamer |
| Clementia Pharmaceuticals | Sponsored Research Agreement | David J Goldhamer |

Clementia Pharmaceuticals collaborated on study design and interpretation of data related to palovarotene effects on heterotopic ossification. Funders did not participate in data collection or the decision to submit the work for publication.

### Author contributions

John B Lees-Shepard, Conceptualization, Data curation, Formal analysis, Investigation, Writing—original draft, Project administration, Writing—review and editing; Sarah-Anne E Nicholas, Data curation, Formal analysis, Validation, Investigation, Writing—review and editing; Sean J Stoessel, Formal analysis, Investigation, Writing—review and editing; Parvathi M Devarakonda, Michael J Schneider, Investigation, Writing—review and editing; Masakazu Yamamoto, Resources, Methodology, Writing—review and editing; David J Goldhamer, Conceptualization, Formal analysis, Supervision, Funding acquisition, Validation, Methodology, Writing—original draft, Project administration, Writing—review and editing

### Author ORCIDs

John B Lees-Shepard http://orcid.org/0000-0002-1275-5799

David J Goldhamer http://orcid.org/0000-0003-4605-8921

### Ethics

Animal experimentation: This study strictly followed the recommendations in the Guide for the Care and Use of Laboratory Animals. All experiments were approved by the University of Connecticut Animal Care and Use Committee (IACUC), and are covered under protocol #A17-015.

### Decision letter and Author response

Decision letter https://doi.org/10.7554/eLife.40814.020

Author response https://doi.org/10.7554/eLife.40814.021

## Additional files

### Supplementary files

• Transparent reporting form

DOI: https://doi.org/10.7554/eLife.40814.018

### Data availability

All data generated or analysed during this study are included in the manuscript and supporting files.

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
