## [Decision Letter]

Congratulations, we are pleased to inform you that your article, "Palovarotene reduces heterotopic ossification in juvenile FOP mice but exhibits pronounced skeletal toxicity", has been accepted for publication in *eLife*.

Your article has been reviewed by three peer reviewers, and the evaluation has been overseen by a Reviewing Editor and Mark McCarthy as the Senior Editor. The reviewers have opted to remain anonymous.

There will be significant interest in this well written paper which is likely to affect further clinical trials in FOP as well solidifying the pathogenic mechanisms of this disease. There are a few minor points that the authors need to address in the Discussion even with the acceptance. The authors should comment on the variable response in dose of palovarotene, the cause of death of juvenile WT mice, and speculate how PVO might enhance cartilage growth. We suggest you place the statistical methods in a separate section of the manuscript and note if the means are represented as SEM or SD. These edits will not affect the decision to publish this paper.

*Reviewer #1:*

In this manuscript Lees Shephard and colleagues demonstrate the in vitro and in vivo effects of palovarotene on a new animal model of spontaneous HO. Their work is based on the hypothesis that activation of the activin A receptor in Pdgfrα FOPs results in spontaneous HO and ultimately FOP. Recently published work by this same group introduced the mouse model which recapitulates spontaneous HO while these studies examine the efficacy of palovarotene vs. an activin A blocking antibody in the treatment of spontaneous HO. Several results highlight the significance of this work including:

1) The relationship of FAP number to disease progression and further characterization of their mouse model, both treated and untreated – i.e. natural history studies;

2) The difference in FAP expansion in treated cells between palovarotene and an activin A blocking antibody;

3) The first head to head comparison of two treatments (activin A antibody vs. palovarotene) in a spontaneous HO model;

4) The synovial overgrowth and joint disruption and long bone growth ablation noted with palovarotene treatment;

Hence the work breaks new ground and is well written. As such it should add considerably to our knowledge base when considering treatment options in this progressively fatal disease, particularly since phase II trials are ongoing with both treatments.

There are a few concerns which need further elaboration:

1) It appears the jaw overgrowth is the cause of death in the treated HO mice; were necropsies done on those mice to establish other off-target effects?

2) It would be important for the reader to know how the doses of palovarotene were established and how that compares to ongoing human trials;

3) In that same vein, it appears as if in both in vitro and in vivo studies, there is not a consistent dose response to palovarotene, particularly in respect to amount of HO over the natural history; can the authors elaborate?

4) Why did palovarotene reduce survival in juvenile WT mice? Were necropsies performed? Also, please show body weight data in treated WT mice.

5) In the transplantation study with SCID mice, why wasn’t the lower dose of palovarotene used?

6) A fundamental question underlying the paper is whether FAP treatment with palovarotene reduces cell survival or proliferation; further data would be helpful in that regard.

*Reviewer #2:*

The authors employ a previously described developmental/juvenile model of FOP that uses Pdgfrα lineage-specific activation of an *Acvr1^R206H^* knock-in allele, demonstrating all known HO phenotypes of FOP, including chostochondral, articular and periarticular, and intramuscular heterotopic ossification, as well as osteochondromas not previously found in mouse models. This developmental model is associated with significant morbidity attributed to poor feeding as a result of jaw HO, resulting in markedly reduced survival evident by P42. This developmental model provides the opportunity for testing various interventions during a window of development (P14 to P42) that corresponds roughly to ages during which FOP manifests in humans, and to examine the potential impact of these treatments on orthotopic skeletal growth. Two treatments are evaluated, an activin A neutralizing antibody and an RAR-gamma agonist, both approaches currently being tested in humans with FOP.

In addition to the developmental model, the authors use an FAP transplantation approach, showing importantly that HO is proportional to the numbers of transplanted and engrafted *ACVR1* mutant FAPs. Moreover, *Acvr1^R206H^* FAPs exhibits enhanced engraftment and proliferation kinetics in comparison to wild-type FAPs, consistent with a pro-proliferative phenotype conferred by *Acvr1^R206H^*. In testing the impact of ex vivo and in vivo modulation of activin, or treatment with RAR-gamma agonist, they found that anti-activin therapy essentially normalizes the engraftment and proliferation of mutant FAPs to nearly that of wild-type FAPs. In contrast, treatment with PVO reduces the survival/proliferation of engrafted mutant FAPs, but does not normalize them to the numbers of WT FAPs.

In the developmental model, treatment with PVO appeared to improve overall HO, and was particularly effective in reducing spontaneous HO at the ankle joints. However, treatment with PVO at both doses exacerbated HO formation in the knees, and possibly the wrist, hip, shoulder joints, and cervical and lumbar spine. Treatment with PVO at both doses led to the formation of abnormal cartilaginous growths in articular cartilage, particularly within the costochondral, shoulder, wrist and knee joints. The exacerbation of HO by PVO in a developmental model of FOP raises important questions about the mechanism of PVO, previously presented as an anti-chondrogenic molecule, but enhancing articular cartilage formation in this model.

With respect to the developmental effects of *Acvr1^R206H^* on orthotopic skeletal growth, expression of *Acvr1^R206H^* in Pdgfrα+ lineages led to a decrease in tibial length but with generally preserved growth plate structure, representing a slight modification of observations by Chakkalakal et al. 2016 in which decreased long bone length was observed with Prrx1+ specific expression of a similar *Acvr1^R206H^* knock-in allele but with altered epiphyseal and hypertrophic zone thicknesses, all of which were partially normalized by PVO treatment in that study. In the current study, treatment with PVO was associated with essentially absent growth plates or findings suggestive of growth plate trauma. Moreover, treatment with PVO was associated with further shortened body length, spine lengths, tibial lengths and skull lengths in treated versus untreated PDGFRα-Cre; *Acvr1^R206H^*/+ animals, as might be expected from a potent retinoid administered during skeletal development. This result contrasts with Chakkalakal et al. in which PVO treatment of Prrx1-Cre; *Acvr1^R206H^*mice appeared to improve long bone length with partial normalization of growth plate structure. On the other hand, the current observations are more consistent with those of Inubushi et al. (2018), in which it was found that PVO treatment of mice without *ACVR1* mutations starting at P14 was associated with decreased long bone length and altered growth plate structure. In the Inubushi paper starting PVO at P21 was associated with less impact on skeletal development. In light of the prior observations, the authors should comment why the P14-P42 window of treatment was chosen, if it was shown to be optimal as compared to delayed treatment windows, and whether or not delaying treatment to P21 might still exhibit reasonable efficacy without causing toxicity?

These current findings have tremendous relevance for current pediatric clinical trials investigating palovarotene for the treatment of FOP. The authors clearly show that doses of PVO (0.75 - 1.5 mg/kg/d IP) required for preventing developmental HO phenotypes in this condition are associated with long bone shortening and dissolution of growth plates, yet also have the potential to create cartilage malformations at sites of articular cartilage and synovial cartilage and paradoxically exacerbate HO at these sites. The mechanism of paradoxically increased cartilage formation and HO following PVO treatment raises questions about whether or not the mechanisms of PVO's effects are matched to the pathophysiology of FOP, and whether this is an ideal therapy for FOP. Could the authors speculate on how PVO might enhance cartilage formation, and enhance HO at some sites while preventing it at others?

In addition to the observed counterproductive or toxic effects of juvenile PVO therapy observed upon long bone length and growth plate morphology/maintenance, one wonders if the relative failure of PVO to contain mutant *ACVR1* FAP proliferation in engraftment studies might have additional untoward effects besides the inadequate control of HO? For example, in the last paragraph of the Discussion the authors discuss that the failure to control the initial expansion of the FAPs is associated with muscle fibrosis following injury. The current data (Figure 6) demonstrates that anti-activin almost normalizes mutant *ACVR1* FAP proliferation and engraftment to that of wild-type, whereas the response even with the higher dose of PVO is enhanced compared to wild-type cells. Could the authors comment on the possibility that PVO might lead to excessive muscle fibrosis or fatty infiltration? Could the muscle tissues of mice from these implantation experiments in Figure 6 be used to assess this question, rather than leaving this open for speculation? The answer may be important, as satisfactory function of the residual muscle, even after successful abrogation of HO, may depend upon a normal muscle injury-healing response, and might be impaired if wound-healing functions of critical populations such as FAPs become impaired or dysregulated as a result of anti-HO therapies. The muscle crush/FAP engraftment model with or without PVO treatment might be advantageous for assessing this, while an analysis of the P42 mutants treated with or without PVO could reveal any other important effects of systemic treatment on muscle during post-natal growth.

*Reviewer #3:*

The manuscript "Palovarotene reduces heterotopic ossification in juvenile FOP mice but exhibits pronounced skeletal toxicity", expands our current understanding of the pathological progression of FOP, the role of FAPs during this progression and the therapeutic potential of 2 differing modes of intervention on dampening FOP pathogenesis. This new information is clearly and carefully presented. The manuscript also reports an unexpected and worrisome finding, that palovarotene treatment has profound negative effects on the skeleton of juvenile mice, a stark contrast to the effects of anti-activin-A treatment that appear to be beneficial to FOP carrying mice and innocuous to WT mice. While only small cohorts of mice were examined using only one specific dosing regime, data are quite convincing and suggest that the age of patients at the start of FOP therapy must be carefully considered prior to treatment as to prevent deleterious actions of palovarotene in the juvenile skeleton.

Additionally, establishing upper limits to drug administration as a function of skeletal age should also be considered. As both anti-activin-A and palovarotene are currently in use in clinical trials for FOP, this study is topical and of strong clinical significance.

My only substantive concern is the amount of variability in efficacy of treatment vs. toxicity that may occur with each individual dosing regime in patients of different ages. While it seems unlikely that the strong negative effects observed with palovarotene will only occur with this dosing schedule in mice of this age, the very compact nature of this study may not allow the authors to generalize about palovarotene treatment in juvenile patients with FOP.

---

## [Author Response]

There will be significant interest in this well written paper which is likely to affect further clinical trials in FOP as well solidifying the pathogenic mechanisms of this disease. There are a few minor points that the authors need to address in the Discussion even with the acceptance. The authors should comment on the variable response in dose of palovarotene, the cause of death of juvenile WT mice, and speculate how PVO might enhance cartilage growth. We suggest you place the statistical methods in a separate section of the manuscript and note if the means are represented as SEM or SD. These edits will not affect the decision to publish this paper.

We were asked to address a few minor points raised in the decision letter.

1) The variable response in dose of palovarotene: In FOP mice, the efficacy and toxicity of palovarotene at the two doses tested (0.735 mg/kg and 1.47 mg/kg) were comparable in most respects and the details were noted in the Results section of the original submission. As relates to efficacy, we have now added a concluding sentence to the relevant paragraph of the Results that specifically notes the apparent limit of palovarotene efficacy in this FOP mouse model. Without a complete dose-response study, we feel it is appropriate to describe similarities and minor differences in dose effects in the Results, without elaborating further in the Discussion.

2) The cause of death of juvenile WT mice treated with palovarotene: We do not know the reason for reduced survival of WT mice (or of FOP mice that did not exhibit heterotopic ossification of the jaw) treated with palovarotene. We now explicitly state in the Results that the reason for reduced survival of palovarotene-treated mice is unknown.

3) Speculate how palovarotene might enhance cartilage growth: In the Results section, we now note the precedent for pro-proliferative effects of retinoids in certain experimental settings.

4) Placement of statistical methods in a separate section: We have now added a section on statistical methods to the Materials and methods section.

In the Materials and methods section, we now include comments concerning the rationale for the doses of palovarotene chosen. This is in response to specific reviewer comments that were not included in the summary review.